# Mechanisms of Resistance to Second-Generation Antiandrogen Therapy for Prostate Cancer: Actual Knowledge and Perspectives

**DOI:** 10.3390/medsci10020025

**Published:** 2022-04-28

**Authors:** Francesco Pinto, Francesco Dibitetto, Mauro Ragonese, Pierfrancesco Bassi

**Affiliations:** Urology Department, Fondazione Policlinico Universitario “A. Gemelli” IRCCS, Catholic University of the Sacred Heart, 00168 Rome, Italy; francescodibitetto@gmail.com (F.D.); mauroragonese@yahoo.it (M.R.); bassipf@gmail.com (P.B.)

**Keywords:** prostatecancer, abiraterone, enzalutamide, apalutamide, darolutamide, castrationresistance, CRPC

## Abstract

Prostate cancer therapy for locally advanced and metastatic diseases includes androgen deprivation therapy (ADT). Second-generation antiandrogens have a role in castration-resistant prostate cancer. Nevertheless, some patients do not respond to this therapy, and eventually all the patients became resistant. This is due to modifications to intracellular signaling pathways, genomic alteration, cytokines production, metabolic switches, constitutional receptor activation, overexpression of some proteins, and regulation of gene expression. The aim of this review is to define the most important mechanisms that drive this resistance and the newest discoveries in this field, specifically for enzalutamide and abiraterone, with potential implications for future therapeutic targets. Furthermore, apalutamide and darolutamide share some resistance mechanisms with abiraterone and enzalutamide and could be useful in some resistance settings.

## 1. Introduction

Prostate cancer is the second most common neoplasm in men with about 1.1 million new diagnoses worldwide in 2012 and the fifth most lethal [1]. Therapeutic strategies for localized disease include active surveillance, radical prostatectomy, or radiotherapy. Androgen deprivation therapy (ADT) has a role in locally advanced and metastatic disease. Despite an initial therapeutic efficiency of 80–90% [2], several patients treated with ADT will develop castration-resistant prostate cancer [3], defined as castrate serum testosterone < 50 ng/dL or 1.7 nmol/L with either biochemical progression (three consecutive rises in PSA at least one week apart, resulting in two 50% increases over the nadir, and a PSA > 2 ng/mL) or radiological progression (either two or more new bone lesions on bone scan or a soft tissue lesion) [4].

In recent literature, androgen receptor (AR) amplifications, mutations, expression of splice variants independent from the ligand, intratumoral androgen production, and other biochemical and biological mechanisms represent the principal mechanism of action in primary ADT resistance [5]. These mechanisms led to new compounds with different mechanisms of action that overcome these types of resistances, such as abiraterone acetate, enzalutamide, apalutamide, and darolutamide, the second-generation antiandrogens [6]. To date, second-generation antiandrogens are approved for treatment of nonmetastatic castration-resistant prostate cancer (CRPC) and metastatic CRPC (only abiraterone and enzalutamide) [7,8]. However, about 20–40% of CRPC patients have no response to abiraterone and enzalutamide [9,10] and even if the patient showed an initial response to enzalutamide or abiraterone, the rates of patients acquiring secondary resistance are high [11]. Many mechanisms have been identified to explain acquired and naïve resistance to these agents.

Herein, we propose a literature review on mechanisms of resistance to second-generation antiandrogen therapy for prostate cancer and new biochemical knots for applying new therapeutic strategies.

## 2. Abiraterone

Abiraterone is an orally administered small molecule that irreversibly inhibits the products of the cytochrome P450 family 17 (CYP17) gene (including both 17,20 lyase and 17-alpha-hydroxylase). It blocks the synthesis of androgens in the tumor as well as in the testes and adrenal glands. Altered steroidogenesis was discussed as one mechanism by which CRPC develops. Adrenal androgen levels are not affected by ADT. Dehydroepiandrosterone (DHEA) and its sulfated form (DHEA-S) are converted to dihydrotestosterone (DHT) via a backdoor pathway [12].

The molecules are converted to androstenedione (AD) either in the prostate or adrenal gland by 3β-hydroxysteroid dehydrogenase (3βHSD), encoded by HSD3B. There are two isoforms, 3βHSD1 in the prostate and peripheral tissues, and 3βHSD2 in the adrenal gland. The conversion from AD to DHT, in the absence of ADT, goes through testosterone as an intermediary and requires 17βHSD3, AKR1C3 and steroid-5α-reductase. However, in the presence of ADT, this sequence can be reversed, leading to 5α-AD (5α-androstenedione) that acts as an intermediary agent, bypassing testosterone completely. This alternative pathway, known as the “5α-dione” pathway, has been demonstrated to predominate in CRPC. So, abiraterone-naive CRPC cells use the 5α-dione pathway to produce intra-tumoral DHT but are still dependent on adrenal androgens. By irreversibly inhibiting this critical upstream enzyme in the steroidogenesis pathway, abiraterone effectively causes a significant decrease in intratumoral androgen levels by preventing production of adrenal androgens. However, despite its effectiveness in inhibiting the steroidogenesis pathway, abiraterone’s effect is incomplete. Overexpression or mutations of CYP17A1 may also contribute to abiraterone resistance. HSD3B1 (1245C) mutation contributing to progression to CRPC has also been found in abiraterone-resistant xenograft models [13]. Mostaghel et al. demonstrated that in abiraterone-treated prostate cancer cells there is overexpression of the isoform CYP17A1, as well as other enzymes involved in steroidogenesis, such as AKR1C3 and HSD17B3 [14]. AKR1C3 facilitates the conversion of androstenedione and 5 α-androstenedione (5 α-dione) that are weak androgens to the more effective testosterone and DHT. It is not only enzyme mutations that can lead to abiraterone resistance. Another important role in resistance to abiraterone in prostate cancer cells is played by androgen accumulation, such as pregnenolone and progesterone. The progesterone receptor (PR) has been found overexpressed in mCRPC [15]. In addition, PR high density is an independent poor prognostic factor [16]. The PR and AR have 88% sequence homology in the ligand-binding domain and share common response elements. Therefore, the accumulation of progesterone could lead to transcription of androgen-dependent genes. A novel therapeutic target could be the inhibition of PR caused by onapristone, a progesterone receptor antagonist that is currently under evaluation in CPRC that has progressed after abiraterone, enzalutamide, or two lines of chemotherapy: preliminary data has shown this is safe and feasible [17].

When CYP17A1 is inhibited, 3βHSD is the main enzyme involved in steroid synthesis. It is responsible for synthesis of progesterone from pregnenolone, androstenedione from DHEA, and testosterone from androstenediol. The androgen receptor isoform encoded by splice variant 7 lacks the ligand-binding domain, which is the target of enzalutamide and abiraterone. This mutation leads to constitutive activation of the receptor as a transcription factor. Antonarakis et al. hypothesized that ARV-7 detection in circulating tumor cells from patients with CRPC may be associated with resistance to enzalutamide and abiraterone [18]. ErbB2 also has been proposed as an actor in resistance to abiraterone in CRPC. Studies in cell line and xenograft models have indicated that increased ErbB2 signaling can contribute to the restoration of AR activity and tumor growth in CRPC [19,20,21]. The physiologic significance of ErbB2 in CRPC is supported by studies showing increased ErbB2 expression or activity in CRPC clinical samples, although this is not a consistent finding, and increased ErbB2 has also been associated with more aggressive primary untreated prostate cancer. However, clinical trials of ErbB2-targeted therapies have not shown efficacy in prostate cancer patients prior to androgen deprivation therapy or in CRPC. Gao et al. showed that ErbB2 signaling was elevated in residual tumors following abiraterone treatment in a subset of patients and was associated with higher nuclear AR expression. ErbB2 and subsequent activation of the PI3K/AKT signaling stabilizes AR protein. Furthermore, concomitantly treating CRPC cells with the abiraterone and ErbB2 inhibitor, lapatinib, blocked AR reactivation and suppressed tumor progression [22]. Annala et al. randomized 202 patients with treatment-naïve mCRPC to abiraterone or enzalutamide and performed whole-exome and deep-targeted gene sequencing of plasma cell-free DNA prior to therapy. They found genomic alterations that were strongly correlated with rapid resistance to abiraterone or enzalutamide therapy, such as somatic arrangements in TP53 or defects in BRCA2 and ATM. AR gene structural rearrangements truncating the ligand-binding domain were identified in patients with primary resistance [23].

Mechanisms of resistance to abiraterone are summarized in Table 1.

## 3. Enzalutamide

Enzalutamide is an androgen receptor signaling inhibitor without agonistic activity. It suppresses the nuclear translocation of active androgen receptors to prevent recruitment of androgen response elements. This leads to cellular apoptosis and inhibition of CRPC cell proliferation. Despite his efficacy, eventually all of the enzalutamide-treated patients became resistant. This could be related to different mechanisms. One of these is the androgen receptor reactivation. Yuan et al. observed that AR is highly expressed and transcriptionally active in CRPC despite enzalutamide treatment [24]. AR-persistent activation could be secondary to somatic mutations, such as the amplification of an acquired androgen receptor enhancer located at 650-kb centromere to the AR [19]. Another study [25] found that galectin-3, a member of the animal lectin family, significantly inhibited the therapeutic effect of enzalutamide by increasing the expression of several androgen receptor target genes, such as kallikrein-related peptidase 3 (KLK3) and transmembrane protease serine 2 (TMPRSS2). Such enhancement of androgen receptor transcriptional activity and expression of androgen receptor-related genes can contribute to enzalutamide resistance. Androgen hormone potentiation could also participate. Liu et al. demonstrated that AKR1C3, a critical enzyme in the biological pathways that lead to the production of testosterone and DHT from weak androgens (androstenedione and 5 α-androstenedione), is upregulated in enzalutamide-resistance prostate cancer cells and that indomethacin, an inhibitor of AKR1C3 activity, could overcome this resistance, reducing cell proliferation [26]. Another mechanism is the inhibition of AR-protein degradation by preventing E3 ligase-mediated ubiquitination. This is due to the binding of the AR by a polycomb group protein, B lymphoma Moloney murine leukemia virus insertion region 1 (BMI1). In an enzalutamide-resistant xenograft model, it was found that the BMI1 inhibitor significantly decreased the development of enzalutamide-resistant CRPC [27]. Another cellular pathway involved in AR activation is c-Myc. One study showed induction of c-Myc’s expression by erythropoietin-producing human hepatocellular (Eph) receptors, which is a key factor for enzalutamide resistance [28].

Enzalutamide resistance can also be achieved by AR splice variants, as discussed for abiraterone. Antonarakis et al. demonstrated a significant overexpression of AR–V7 in enzalutamide-treated patients. TK activated cdc42-associated kinase 1 (ACK1) (TNK2) could work in an epigenetic circuit, contributing to the overexpression of AR-V7 and leading to enzalutamide resistance [29]. Another possible target in AR-V7 amplification could be the vasopressin1A (V1A) receptor (AVPR1A) [30]. AR-V7 overexpression could also be related to ubiquitin E3 ligase proteasome degradation: the interaction between heat shock protein family member HSP70 and functional E3 ubiquitin ligase STIP1 homology and U-box containing protein 1 (STUB1) was required for androgen receptor variant 7 proteostasis. Suppression of HSP70 inhibited tumor growth and enzalutamide resistance by decreasing AR-V7 levels [31]. Inhibition of ubiquitination is also achieved by activation of anti-apoptotic B cell lymphoma-2 (BCL2) protein [32].

CRPC patients could be resistant to enzalutamide because of androgen receptor mutations. The most frequent mutation site is the carboxy-terminal ligand-binding domain [33]. For instance, the androgen receptor F876L mutation was previously found to convert the androgen receptor antagonist enzalutamide to an androgen receptor agonist [34]. The double mutation F877L/T878A converts enzalutamide into a fully functional agonist. This obstacle can be overcome by another androgen receptor inhibitor, darolutamide, which works against AR-mutated variants [35].

Glucocorticoid and androgen receptors have overlapped intracellular pathways, and this is the reason why AR inhibitors such as enzalutamide, via a negative feedback mechanism, prevent inhibition of glucocorticoid receptors. This leads to a re-expression of approximately 50% of the androgen receptor-responsive genes. Therefore, glucocorticoid receptor (GR) agonist dexamethasone can determine enzalutamide resistance and the GR antagonist can restore enzalutamide sensitivity [36]. The challenge in exploring therapeutic strategies related to this aspect is that complete blocking of GR is incompatible with life.

Another cellular pathway involved in enzalutamide resistance in CRPC is the Wnt signaling. The Wnt family consists of 19 cysteines and secreted lipo-glycoproteins that regulate stem cell self-renewal, proliferation, migration, and differentiation during organ development. Wnt can bind to different proteins to activate different signaling pathways, but the most studied is the Wnt/β-Catenin affecting cell proliferation, differentiation and transition from epithelial-to-mesenchymal in prostate cancer. In enzalutamide-resistant CRPC cells, there is an upregulation of stem-like genes upon the activation of the Wnt/β-Catenin pathway that reactivates androgen receptor signaling [37]. Another Wnt-activated pathway involves the Ca^2+^/Calmodulin-dependent kinase type 2 [38]. Wnt mutations are present in 10–20% of advanced prostate cancer, and these mutations could affect the response to enzalutamide therapy. A recent study found a poor response to abiraterone or enzalutamide in metastatic CRPC with WNT pathway mutations [39].

Even metabolic changes can participate in CRPC’s resistance to enzalutamide. In neoplastic cells there is a shift towards aerobic glycolysis (Warburg effect), and particularly in CRPC cells, there is an upregulation of glucose transporters (GLUT). GLUT1 expression is associated with a poor prognosis [40]. The GLUT1 gene promoter directly binds to AR, promoting GLUT1 transcription. While the Warburg effect is common in all cancer cells, one unique feature of prostate cancer cells is the hyperactivation of the citric acid cycle. In particular, during the evolution and progression of CRPC, there is a constant increment of malate dehydrogenase-2 (MDH2). Latonen et al. showed that the knockdown of MDH2 can decrease cell proliferation and improve drug sensitivity [41].

Another metabolic pathway that is related to enzalutamide resistance is autophagy through hyperactivation of the mammalian target of rapamycin (mTOR). mTOR is inhibited when, in absence of nutrients or growth factors or under hypoxia, autophagy is activated. Autophagy contributes to enzalutamide resistance by activation of AMPK and inhibition of the mTOR pathway [42].

As discussed for abiraterone, also for enzalutamide resistance there is a role played by the inhibition of apoptosis, especially by BCL2 proteins [43] and inhibitors of apoptosis (IAP) family [44]. In particular, IAP family member BIRC6 is upregulated in enzalutamide resistance [45]. A novel therapeutic target could be a combination between enzalutamide and IAP inhibitors [46].

Another mechanism that drives enzalutamide resistance in CRPC is cell lineage plasticity and phenotypic switch. As said before for the Wnt/β-Catenin complex, the epithelial-to-mesenchymal transition is significantly enhanced by enzalutamide treatment [47], and one of the most important drivers in this transition is TGF-β [48]. In particular, Pal et al. demonstrated that TGF-β pathway regulators, SMAD family member 3 (SMAD 3) and cyclin D1, participate in enzalutamide resistance [49]. It has been demonstrated by Svensson et al. that treatment with enzalutamide may reduce the expression of the repressor element 1 silencing transcription factor (REST), a mediator of AR action on gene repression that may be responsible for neuroendocrine differentiation [50]. Thus, in comparison to a single enzalutamide treatment, adding the TGF-β receptor I kinase inhibitor galunisertib (LY2157299) could enhance the efficacy of enzalutamide in inhibiting castration-resistant prostate cancer xenograft tumor growth and metastasis [51].

Prostate cancer cells express autocrine IL-6, which activates the JAK/STAT3 pathway. IL-6/STAT3 is associated with enzalutamide-resistant CRPC [52]. Other cytokines involved in CRPC progression are IL-23 and C–X–C motif chemokine ligand 12 (CXCL12). CXCL12 and its ligand, CXCR4, are overexpressed in prostate cancer cells. The intracellular CXCL12γ isoform has been found to be expressed in circulating tumor cells of mCRPC and prostate cancer cells with neuroendocrine phenotype; its overexpression could lead to the activation of intracellular pathways associated with cancer progression and drug resistance [53].

Regulation of all these endogenous processes is achieved via different mechanisms. One of them involves microRNAs (miRNAs). miRNAs are short (19–25 nucleotides) endogenous RNA molecules that regulate gene expression after transcription. In CRPC, there are several miRNAs controlling some of the pathways that lead to enzalutamide resistance and cancer progression, such as miR-346, miR-361–3p, and miR-197, that augment androgen receptor activity [54]. miRNA-194 could drive the transdifferentiation of prostate cancer cells into neuroendocrine-like cells by targeting FOXA1 [55]. Some other miRNAs can sensitize response to enzalutamide in CRPC and suppress pathologic biochemical pathways: miR-644a, for example, contributes to the blocking of the the epithelial-to-mesenchymal transition and suppresses the Warburg effect, therefore possibly becoming an auxiliary agent in the treatment of enzalutamide resistance [56].

Other pathways associated with enzalutamide resistance include the NOTCH signaling pathway, independent from AR [57], and the cholesterol synthesis pathway involving 3-hydroxy3-methylglutaryl-CoA reductase (HMGCR), an enzyme found to be upregulated in enzalutamide-resistant CRPC [58]. Khort at. al identified three candidate genes, acetyl-CoA acetyltransferase 1, mixed-lineage kinase 3 (MLK3), and proteasome 26S subunit, non-ATPase 12 (PSMD12), validated as supporters of enzalutamide resistance in vitro, although in vivo behavior needs to be verified [59].

Mechanisms of resistance to enzalutamide are summarized in Table 2.

## 4. Darolutamide and Apalutamide

Darolutamide is a novel AR antagonist that has the capacity of binding both wild-type and mutated AR, trying to overcome enzalutamide resistance. Moreover, compared to apalutamide and enzalutamide, it has a low penetration of the brain blood barrier and a low binding affinity for the gamma-aminobutyric acid (GABA) type A receptor. Borgman et al. evaluated the efficacy of darolutamide in inhibiting cell growth and tumor progression in enzalutamide-resistant CRPC and in mutated AR patients previously treated with abiraterone, enzalutamide, or bicalutamide. Darolutamide significantly inhibited cell growth and AR transcription activity in vitro while decreased tumor volume and serum prostate-specific antigen levels in vivo. In addition, it had a significative inhibition of transcriptional activity of AR-mutated variants, such as F877L, F877L/T878A and T878G, that, as said before, transform enzalutamide into an agonist (Borgmann et al.). 

Apalutamide is a new nonsteroidal AR antagonist that binds directly to the ligand-binding domain of the AR and prevents its translocation and its related transcriptional pathway. Apalutamide has demonstrated an improvement in overall survival (OS) compared to placebo in nonmetastatic CRPC and a longer metastasis-free survival [60].

Currently, there are not many studies that evaluate darolutamide’s and apalutamide’s unique resistance mechanisms. It has been demonstrated that there is a cross-resistance mechanism between darolutamide, apalutamide, enzalutamide, and abiraterone.

## 5. Cross-Resistance

It is interesting to highlight that those patients who receive abiraterone or enzalutamide as the first line of therapy have a 15–30% response rate to the alternative agents as second-line CRPC treatment. This underlines that there clearly exists a cross-resistance between enzalutamide and abiraterone. Resistance to second-line therapy takes about 3–6 months to develop, significantly shortening the duration of benefits from this treatment by at least 50% compared with that of the first-line [61].

As suggested before, some biological mechanisms of resistance to these drugs overlap. Upregulation of CYP17 occurs both in resistance to enzalutamide and abiraterone. A second mechanism that leads to cross-resistance is the upregulation of the AR, due to the amplification of the gene or the overexpression of the protein. Another mechanism common to the two drugs is the emergence of AR splice variants (for example ARV-7), in which abnormal splicing of the AR messenger RNA (mRNA) leads to the formation of a truncated AR protein that is constitutively active without the necessity of a ligand [62].

What is interesting is that cross-resistance can also involve darolutamide and apalutamide, in addition to enzalutamide and abiraterone. Zhao et al. demonstrated that enzalutamide- and abiraterone-resistant prostate cancer cells are also resistant to apalutamide and darolutamide. In particular, the presence of ARV-7 conferred resistance to enzalutamide and apalutamide [63].

Moreover, the emergence of resistance to antiandrogens such as enzalutamide determine the potential for accelerating metastatic disease, as Simon et al. discovered. This implies that the treatment must not be interrupted, even when drug resistance has been achieved [64].

## 6. Future Perspectives

New therapeutic approaches are being studied for CRPC. One randomized trial evaluated the efficacy of Olaparib, a poly(adenosine diphosphate–ribose) polymerase (PARP) inhibitor in mCRPC patients who had a disease progression while receiving a new generation antiandrogen (i.e., enzalutamide or abiraterone). All these patients had alterations in predetermined genes related to homologous combination repair of DNA and were divided into two cohorts, based on the mutations found on tumor tissue: cohort A had at least one alteration in BRCA1, BRCA2 or ATM, cohort B had alterations on 12 other prespecified genes. Patients in cohort A treated with Olaparib had a significantly longer overall survival (18.5 vs. 15.1 months) compared to the control group (that received enzalutamide or abiraterone/prednisone). A significant benefit for Olaparib for imaging-based progression-free survival was seen in the overall population [65].

AKR1C3 could also be a potential therapeutic target for CRPC: Endo et al. found a potent novel AKR1C3 inhibitor, N-(4-fluorophenyl)-8-hydroxy-2-imino-2H-chromene-3-carboxamide, that had the capacity of suppressing the proliferation of prostate cancer cells and augmenting apoptotic cell death induced by enzalutamide and abiraterone [66]. 

The point mutation of androgen receptor AR^F876L^ is stimulated instead of inhibited by enzalutamide. Wu et al. found a novel antagonist of the androgen receptor, JJ-450, capable of inhibiting both wild type AR and mutated one AR^F876L^ [67], but in vivo studies are necessary to understand the potential of this molecule in the therapeutic approach to CRPC. The most important AR mutation, the ARV-7 splice variant, could be targeted by the N-terminal domain antagonists EPI-002 (Ralaniten) and EPI-7170 (that has 8–9 times improved potency that ralaniten). A combination of EPI-7170 and enzalutamide resulted in the inhibition of the proliferation of enzalutamide-resistant prostate cancer cells in vitro with a synergistic action. Moreover, this drug enhanced the effect of enzalutamide in enzalutamide-resistant CRPC preclinical models [68,69].

Targeting the c-Myc pathway via the inhibition of 5-lipoxygenase (5Lox) kills the enzalutamide-resistant prostate cancer cells in vitro while not affecting normal cells in the same experimental conditions [70].

Other potential targets under investigation are STAT5 and carnitine palmitoyltransferase 1B (CPT1B) [71,72].

## 7. Conclusions 

Second generation antiandrogens are newer therapeutic weapons against CRPC. They offer benefits in terms of a longer OS and metastasis-free survival, but eventually all patients become resistant. Understanding the mechanisms behind acquired and naive resistance is important in order to define adjuvant therapies or new therapeutic targets that could extend the time before the appearance of resistance and the overall survival for these patients. There are many important biochemical pathways that prostate cancer cells could modulate in order to achieve resistance to second generation antiandrogens. This modulation is due to different mechanisms: upregulation, overexpression, mutations, metabolic switches, and many more. Not all these patterns are activated in one single CRPC cell and in one single patient, and this makes the therapeutic approach very hard; what can act in one patient could not work in another. This perspective makes it necessary, in our opinion, to find new compounds that can act on different biochemical knots in order to tailor patient care.

The understanding of these new mechanism could lead to a new way of labeling castration-resistant prostate cancer: in the future, it is likely there will be different “phenotypes” in prostate cancer resistant to second-generation antiandrogens defined by the biochemical pathways activated to escape from ADT, and each phenotype could be treated by using different drugs alone or in combination with ADT. Further studies are needed to find out new compounds and evaluate their efficacy in vitro and especially in vivo.

## Figures and Tables

**Table 1 medsci-10-00025-t001:** Mechanisms of resistance to abiraterone.


Activation of the “5α-dione” pathway	Chang et al. [10]
Overexpression of CYP17A1	Mostaghel et al. [11]
Androgen-receptor splice variant 7 (constitutive activity)	Antonarakis et al. [12]
Increased ErbB2 signaling	Gao et al. [16]
Genomic alterations (TP53-BRCA2-ATM-AR)	Annala et al. [17]

**Table 2 medsci-10-00025-t002:** Mechanisms of resistance to enzalutamide.


Androgen receptor (AR) persistent activation	Yuan et al. [18]
AR splice variants	Antonarakis et al. [12], Zhao et al. [24]
AR mutations	Snow et al. [27], Korpal et al. [28]
Glucocorticoids receptor upregulation	Arora et al. [30]
Activation of Wnt pathway	Zhang et al. [31]
Metabolic modifications	Wang et al. [34], Latonen et al. [35], Nguyen et al. [36]
Inhibition of apoptosis	Siddiqui et al. [37], Krajewska et al. [38]
Phenotypic switch	Miao et al. [41], Pal et al. [43]
Cytokine signaling	Canesin et al. [45], Jung et al. [46]
miRNA regulation of gene expression	Fletcher et al. [47], Ebron et al. [49]
Other pathways (NOTCH, HMGCR)	Farah et al. [50], Yuan et al. [51]

## Data Availability

Not applicable.

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
