# Peer review of "Mechanisms of Resistance to Second-Generation Antiandrogen Therapy for Prostate Cancer: Actual Knowledge and Perspectives"

_medsci, 2022, doi:10.3390/medsci10020025_

Round 1

Reviewer 1 Report

Med. Sci. 2022 review 1.31.2022

The manuscript by Pinto et al.

The manuscript is investigating on resistant therapeutic barrier for prostate cancer therapy in 2nd-generation antiandrogen treatment. And the resistance appeared has reduced the treatment efficacy and has become some main problems for treating cancer therapy in CRPC. The authors summarized current drugs, that are Enzalutamide, Darolatamide and Apalutamide with some AR-mutated variants and suggested the future perspectives. It is very important issue. However, the concepts for AR resistance and mechanism were already announced at similar title and concept (Verma et al. Cancer Drug Resist 2020;3:742-61, DOI 10.20517/cdr.2020.45 as attached). And some questions are on of Abstract (Line 15 and 16) and introduction (Line 19 and 20). That is not new (current) concepts and 10 years (2012, line 20) ago reference couldn’t be current guidelines (Line 19). And also, most references should follow the unique font, style and size and some publication year missing.

As a result, the manuscript should be not published to this journal.

Author Response

  • AR resistance’s mechanisms are based on the most recent literature available (such as the other resistance mechanisms). It is consequential that some mechanisms have been already described but the aim of this paper is to summarize the most of the processes we know, the oldest and the newest, in order to provide a comprehensive idea of how it’s established a resistance to second generation antiandrogens.
  • The epidemiologic reference is on the current 2022 EAU guidelines (https://uroweb.org/guidelines/prostate-cancer/chapter/epidemiology-and-aetiology) but, in order to put a newer reference, we took in this reviewed version datas from GLOBOCAN 2020.
  • We updated the font of the references and added the publication year.

Reviewer 2 Report

The manuscript reviews mechanisms of resistance to second generation antiandrogen drugs, in particular abiraterone and enzalutamide. The information presented in the manuscript is valuable. However, the organization and presentation of the manuscript hinder the accessibility to the information. There are no clear structure to the manuscript aside from just simply listing the information in lengthy paragraphs. The tables and references are also poorly made. For a review/synthetic article, the presentation matters significantly. Hence, due solely to the current presentation of the manuscript, I cannot recommend it for publication.

Author Response

  • The aim of our paper was to provide a simple, cohesive and clear presentation of the mechanism of resistance to second generation-antiandrogen, according to current literature. We thought it could be a useful informative tool for a student, a resident or a physician and a good source from which propose ideas for new studies. The nature of this paper is descriptive and its organization and presentation are consequential. We reached your suggestion about the tables and we updated the presentation in a less poor way. However, their content is simply a summary of the resistance mechanisms presented in the paragraph above and it couldn’t be different from that.

Round 2

Reviewer 2 Report

The authors addressed my concerns. The review will provide valuable information for the targeted audience.